# Intra-Cellular Calcium Signaling Pathways (PKC, RAS/RAF/MAPK, PI3K) in Lamina Cribrosa Cells in Glaucoma

**DOI:** 10.3390/jcm10010062

**Published:** 2020-12-26

**Authors:** Mustapha Irnaten, Aisling Duff, Abbot Clark, Colm O’Brien

**Affiliations:** 1Department Ophthalmology, Mater Misericordiae University Hospital, D07 R2WY Dublin, Ireland; cobrien@mater.ie; 2Milton Medical Centre New South Wales, Milton, NSW 2538, Australia; aisling.duff@gmail.com; 3Department Pharmacology & Neuroscience and the North Texas Eye Research Institute, Health Science Center, Fort Worth, TX 76107, USA; abe.clark@unthsc.edu; 4School of Medicine and Medical Science, University College Dublin, D04 V1W8 Dublin, Ireland

**Keywords:** glaucoma, lamina cribrosa, fibrosis, calcium, PKCα, *p*38-MAPK, *p*42/44-MAPK

## Abstract

The lamina cribrosa (LC) is a key site of fibrotic damage in glaucomatous optic neuropathy and the precise mechanisms of LC change remain unclear. Elevated Ca^2+^ is a major driver of fibrosis, and therefore intracellular Ca^2+^ signaling pathways are relevant glaucoma-related mechanisms that need to be studied. Protein kinase C (PKC), mitogen-activated MAPK kinases (*p*38 and *p*42/44-MAPK), and the PI3K/mTOR axis are key Ca^2+^ signal transducers in fibrosis and we therefore investigated their expression and activity in normal and glaucoma cultured LC cells. We show, using Western immune-blotting, that hyposmotic-induced cellular swelling activates PKCα, *p*42/*p*44, and *p*38 MAPKs, the activity is transient and biphasic as it peaks between 2 min and 10 min. The expression and activity of PKCα, *p*38 and *p*42/*p*44-MAPKs are significantly (*p* < 0.05) increased in glaucoma LC cells at basal level, and at different time-points after hyposmotic stretch. We also found elevated mRNA expression of mRNA expression of PI3K, IP3R, mTOR, and CaMKII in glaucoma LC cells. This study has identified abnormalities in multiple calcium signaling pathways (PKCα, MAPK, PI3K) in glaucoma LC cells, which might have significant functional and therapeutic implications in optic nerve head (ONH) fibrosis and cupping in glaucoma.

## 1. Introduction

Glaucoma is an optic neuropathy characterized by optic nerve head (ONH) cupping and pallor and loss of retinal ganglion cells leading to blindness [1]. In the ONH, the connective tissue lamina cribrosa (LC) is a key site of damage in glaucoma. Pathological features include excessive extracellular matrix (ECM) accumulation in the LC area that leads to tissue remodeling and fibrosis [2,3]. The ECM remodeling includes increased deposition of collagens [4], loss of elastin structure [5], and higher transforming growth factor-β2 (TGF-β2) and matrix metalloproteinase-2 (MMP-2) protein levels in human glaucoma LC tissue [6].

Fibrosis is a molecular process defined by the accumulation of excess of ECM which ultimately damages the connective tissue leading to impairment of normal function; it occurs in multiple tissues and organs including heart, kidney, lung, skin, liver [7,8,9], pancreas, intestine, and eye [10,11]. Fibrosis is the result of tissue stiffening caused by the excess deposition of ECM proteins in response to chronic inflammation, provoked by different stimuli such as hypoxia, oxidative stress, mechanical stretch, and growth factors (e.g., TGFβ). These stimuli can promote fibroblasts to proliferate, migrate, and acquire an activated phonotype change into myofibroblasts contributing to further tissue remodeling and subsequently fibrosis [10]. The transition of a normal fibroblast to a myofibroblast cell also involves altered expression of various proteins required in calcium (Ca^2+^) regulation and signaling, including protein kinase C (PKC) and mitogen-activated protein kinase (MAPKs) [12]. Therefore, aberrant elevation of intracellular [Ca^2+^]_i_ levels may lead to cell proliferation and thus contribute to fibrosis and carcinogenesis [13]. It has been shown in cancer cells that the mechanism behind Ca^2+^-induced cell proliferation is caused by the opening of Ca^2+^ channels allowing Ca^2+^ entry, followed by downstream activation of the MAPKs pathway promoting cell proliferation [14].

In glaucoma, both the tissues of the LC and the trabecular meshwork (TM) demonstrate considerable ECM fibrosis [15,16]. Previous work from our group has focused on deciphering the different fibrotic signatures of LC cells in response to glaucomatous change. We have shown that glaucoma LC cells [17], exhibit a hyper-proliferative phenotype, with excessive ECM gene production and associated ECM stiffness, and pro-fibrotic marker expression (e.g., collagen 1A1, periostin and fibronnectin), upon stimulation with TGFβ [18], cyclic stretch [19], and hypoxia [20]. Such cell-associated changes are typical of fibroblast to myofibroblast transition and fibrotic disease progression. Furthermore, we found that LC cells grown on stiff substrates show enhanced expression of αSMA, F-Actin and vinculin [21].

In addition, we previously used hyposmotic-induced cellular swelling (hypotonic-induced stretch) to model glaucoma in LC cells, and we found that both, basal and hyposmotic intracellular Ca^2+^ are raised in glaucoma LC cells [22]. Furthermore, we showed that the Ca^2+^-activated Maxi-K^+^ channels (basal and hyposmotic-induced cell stretch) expression and activity are raised in glaucoma LC cells [23]. More recently, we found that subtypes of the Ca^2+^-permeable, stretch-activated cation channels, transient receptor potential canonical TRPC1 and TRPC6, contribute to the elevated intracellular [Ca^2+^]_i_ found in glaucoma LC cells by reducing the oxidative stress-induced production of ECM genes and LC cell proliferation via a signaling pathway mechanism involving nuclear factor of activated T-cells NFATc3 [24].

It is well known that, in response to mechanical stress and/or stretch stimulus, cells identify and translate a diverse set of signaling events to downstream cellular responses inducing changes in ECM production, apoptosis, and proliferation [25]. It is also known that any change in [Ca^2+^]_i_, ultimately induce changes in its related downstream proteins expression and activity by a wide range mechanisms [26]. The first mechanism is, in response to external stimuli, PKCs bind to Ca^2+^-binding domains, which, once activated, contribute to the activation of the protein’s function. PKCs respond to the second messengers Ca^2+^ and diacylglycerol (DAG) to express their activity at the cell membrane (Figure 1A) [25].

The second mechanism is the Ca^2+^ mediated Ca^2+^-CaM-protein kinase II (CaMKII) activation. This mechanism occurs when levels of intracellular Ca^2+^ are raised. Ca^2+^ rise induces formation of a complex with calmodulin (Ca^2+^-CaM), and prolonged [Ca^2+^]_i_ increase phosphorylates CaMKII and induce its association with the Ca^2+^-CaM complex. A third, mechanism is the Ca^2+^ mediated RAS-RAF-MAPKs activation, including activation of protein kinases *p*38-MAPK and *p*42/44-MAPK.

MAPKs are protein Ser/Thr kinases that sense, identify, and transduce external stimuli, such as receptor tyrosine kinases (RTK), G-protein coupled receptors (GPCR), and growth factor receptors (EGFR) involved in signaling processes.

MAPKs are serine/threonine proteins that are expressed in a wide range of cell types. These kinases are crucial in multiple cellular activities including metabolism, cell differentiation, cell growth and death. Over expression/activity of these kinases can promote numerous signaling pathways including cell proliferation, invasion, and stress responses. The MAPK family consists of different kinases which include MAPKs (MAPK, MAPKK, MAPKKK), external signal-regulated kinases *p*42/44-MAPK (also known as ERK ½), c-Jun N-terminal kinase (JNK), and *p*38-MAPK [27], (Figure 1B). To regulate their downstream target(s), MAPKs phosphorylate their own Ser/Thr residues (auto-phosphorylation). MAPKs activation occurs in a succession of phosphorylations where each MAPK is phosphorylated by an upstream MAPK. Activated MAPKKK induces phosphorylation of an upstream MAPKK, which then, in turn, phosphorylates an upstream MAPK.

The activation of p38-MAPK is started by the binding of a ligand to a receptor tyrosin kinase or a G-protein couple receptor, which in turn, activate small G-protein, Ras. The complex may promote numerous intracellular signaling pathways, especially those contributing to the production of fibrotic regulators [28]. *p*38-MAPK activation results in a series of phosphorylation of its related downstream kinases [29,30]. Over-expression/activity of *p*38-MAPK has been reported in numerous fibrotic organs including pulmonary fibrosis, peritoneal membrane, and cardiac fibrosis [31,32,33].

The PI3K/Akt/mTOR pathway is frequently associated to cellular physiology and patho-physiology, including cell metabolism, differentiation, and proliferation. Dysfunction of the PI3K/Akt/mTOR signaling pathway has been shown in pro-fibrotic diseases including cancer, and pulmonary and cardiac fibrosis. Over expression/activity of one of the PI3K/Akt/mTOR components, frequently promotes metabolism, cell growth, and cell death. Inhibitors of PI3K/Akt/mTOR pathway have been extensively investigated in cancer therapy [34] Therefore, PI3K/Akt/mTOR axis may have a central role in regulating the over-expression of ECM genes and over-proliferation found in glaucoma LC cells.

So far, no study has reported on the link between *p*38-MAPK, *p*42/44-MAPK, and the PI3K/mTOR signaling pathway in glaucoma. Therefore, this study explores the expression of these kinases in glaucoma LC cells, and since this understanding can help clarify the pathological process behind fibrosis in glaucoma, this may provide a roadmap for potential therapeutic interventions to delay or avoid fibrosis initiation LC cells and cupping of the optic nerve in glaucoma.

## 2. Material and Methods

### 2.1. LC Cell LC Culture and Characterization

Human primary cultures of LC cells were obtained from age-matched donors with no history of ocular diseases (controls) and from donors with confirmed glaucoma (Alcon Labs, Fort Worth, TX, USA and from Duke University, Durham, NC, USA). We have previously characterized and described LC cells [24]. Briefly, newly thawed cells were usually stained positively for a fibroblast marker, α-smooth muscle actin (α-SMA), and negatively for an astrocyte marker, glial fibrillary acidic protein (GFAP) and a microglial marker, ionized Ca^2+^ binding adapter molecule 1(Iba1). Cells were cultured at 37 °C in Dulbecco’s Modified Eagle’s–Medium (DMEM) (Sigma, Dublin, Ireland) supplemented with 10% heat inactivated feotal bovin serum (FBS) (vol/vol) and 1% l-glutamine and 1% antibiotics (Sigma, Dublin, Ireland) (penicillin-streptomycin) under humidified air (5% CO_2_). LC cells were cultured in full medium and passaged when they have proliferated to confluence (~80%). Cells of the 2nd to 9th passage were used in the experiments. Before treatment with hyposmotic-induced cellular swelling, LC cells were deprived of serum for 24 h.

### 2.2. Hyposmotic-Induced Cellular Swelling Treatment of Lamina Cribrosa Cells

The effect of glaucoma related stimulus (hyposmotic-induced cellular swelling) on membrane protein kinases including PKCα, p38MAPK, and p42/44MAPK expression and activity was investigated in normal and glaucoma LC cell primary cultures.

The isotonic (isosmotic) bathing solution contained, in mM: 120 NaCl (Sigma, Dublin, Ireland); 6 KCl; 1 MgCl2 (Sigma, Dulin, Ireland); 5.4 HEPES (Sigma, Dublin, Ireland) and 80 D-manitol (Sigma, Dublin, Ireland), pH 7.4 was calibrated with NAOH (Sigma, Dublin, Ireland), with an average osmolarity of 323 mOsm. The hypotonic (hyposmotic) solution was prepared by replacing D-manitol with sucrose from isosmotic solution, pH 7.4, with an average osmolarity of 232 8 mOsm. The osmolarity was measured instantly prior to experiments. For protein expression measurements, normal and glaucoma LC cells were first incubated for at least 15 min in isosmotic bathing solution to obtain a stable baseline, then the hypotonic solution was rapidly applied to create hyposmotic cell membrane stretch. Expression and activity of PKCα, p38, and p42/44-MAPK were measured under isosmotic conditions and after cells were subjected to hyposmotic solution at the time-points of 1, 2, 5, 10, 20, 30 and 60 min.

### 2.3. RNA Isolation, cDNA Synthesis and Quantitative Real-Time RT-PCR

Total cellular RNA was isolated from confluent T75 flasks (ThermoFisher, Dublin, Ireland) containing at least 10^6^ of normal and glaucoma LC cells using Tri-Reagent (Life Technologies, Dublin, Ireland). The first strand complementary DNA (cDNA) was synthesized from 2 µg of total RNA using enhanced avian reverse transcriptase (eAMV), (Sigma, Arklow, Ireland), oligodTs (Sigma, Arklow, Ireland), deoxynucleotides (dNTPs) (Sigma, Arklow, Ireland), and the equivalent primers. The samples containing cDNA were then stored at −20 °C until use.

qRT-PCR was performed on white, flatbottom, 96 well plates (ThermoFisher, Dublin, Ireland) on a Rotorgene 3000 Real-Time PCR Thermocycler (Labortechnik, Wasserburg, Germany) using QuantiTect SYBR Green PCR Master Mix (Qiagen, London, UK). The qRT-PCR mixture contained 2X SYBR Green Master mix, 50 ng od cDNA, 0.5 µM of each primer and PCR-grade water up to total volume of 20 µl. The ribosomal RNA 18S (rRNA 18S) was used as an internal housekeeping control gene. The PCR primer sequences used are as follows:

IP3R 1 forward: 5′-ATTGTCAGCGTGGGTCTGG-3′; reverse: 5′-CAAT CATATACATACAATACAAAACCGAGT-3′, CaMKII: forward: 5′-TTTCCCATCGCCGGAAT-3′; reverse: 5′-GCGTTTGGATGGGTTAATGGT-3′, PI3K: forward: 5′-GGTTGTCTGTCAATCGG TGACTGT-3′, reverse: 5′-GAACTGCAGTGCACCTTTCAAGC-3′, mTOR: forward: 5′-GCTTGATT TGGTTCCCAGGACAGT-3′, reverse: 5′- GTGCTGAGTTTGCTGTACCCATGT-3′, and 18S rRNA: forward: 5′-TGTGCCGCTAGAGGTGAAATT-3′; reverse: 5′-TGGCAAATGCTTTCGCTTT-3′. Each PCR product was analyzed based on the individual cycle threshold, by the 18S rRNA standard curve. All gene expression levels were normalized to 18S rRNA, and data were quantified according to the method used by Livak [35], and presented as the mean ± S.D. All qRT-PCR reactions were performed with a negative control (no template) and repeated at least three times on three biological replicates.

### 2.4. Cell Lysate Preparation and Western Blot Analysis

Normal ang glaucoma LC cells were seeded on 60-mm tissue culture dishes, depleted of serum for 24 h and then treated with isosmotic (isotonic) physiological solution (control) or hyposmotic (hypotonic) solution for various time periods. Following treatment, cells were washed twice with ice-cold phosphate buffer saline (PBS) (Sigma, Ieland)) solution and collected in ice-cold PBS. Cells were centrifuged (1000× *g*, 5 min, 4 °C) and the supernatant was removed. The cells were then lysed in radio immuno-precipitation assay (RIPA) buffer containing protease/phosphatase inhibitor cocktail (Sigma Aldrich, Arklow, Ireland). Following incubation on ice for 10 min, cells were centrifuged (14,000× *g*, 4 °C, 15 min). The supernatant was then collected, and proteins concentration was measured using Bradford assay. Proteins (20 μg/lane) were separated by 10% SDS-PAGE and transferred onto nitrocellulose membranes. The membranes were blocked with 5% fat-free milk in Tris Buffer Saline (Sigma, Dublin, Ireland) supplemented with 0.1% Tween-20 (Sigma, Dublin, Irland) (TBST) for 1 h at room temperature and then incubated overnight at 4 °C with anti-phospho-PKCα (1:1000 dilution, Abcam, Cambridge, UK), anti-phospho-*p*42/*p*44 MAPK (1:1000 dilution, Abcam, Cambridge, UK), and anti-phospho-*p*38 MAPK (1:1000 dilution, Abcam, Cambridge, UK), primary antibodies, respectively. After washing with TBST, the membranes were probed with the corresponding secondary antibodies coupled to horseradish peroxidase for 1 h at room temperature (1:5000 dilution, Abcam, Cambridge, UK). Protein signals were detected with enhanced chemiluminescence (ECL) Western blotting substrate (Fisher Scientific, Dublin, Ireland) and analyzed with Image J software (NIH, LOCI, University of Wisconsin, USA). Membranes were re-probed with anti-total PKCα, anti-total *p*42/44-MAPK and anti-total *p*38-MAPK antibodies and with beta-actin and confirmed equal protein loading in the gels. The experiments were repeated three times on three biological replicates.

### 2.5. Statistical Analysis

Measurement data are expressed as means ± S.E from the corresponding set of experiments. Groups are compared using one-way analyses of variance (ANOVA) for comparison of 3 or more groups, and with a Student’s t-test when comparing 2 groups. The probability values below 0.05 (*p* < 0.05) were considered significant. Calculations were performed using the Origin 7.0 (Origin Lab) software (Origin Lab, Bucks, UK).

## 3. Results

### 3.1. Basal and Hyposmotic-Induced Expression/Activity of PKCα in Normal and Glaucoma LC Cells

The induction of PKCα expression and activity using hyposmotic-induced cellular swelling to model glaucoma has not been reported in LC cells. Here, we performed a time-course study wherein we examined the expression/activity of PKCα in normal and glaucoma LC cells cultured under resting (isotonic) conditions and stimulated with hyposmotic-induced cellular swelling solution. Figure 2 shows that when LC cells are incubated in isotonic physiological solution, the basal expression level of PKCα was significantly greater in glaucoma LC cells (3.08 ± 0.48, p-PKCα levels normalized to total PKCα levels 1 × 10^3^ a.u (arbitrary units)) versus normal LC cells (1.92 ± 0.49, p-PKCα levels normalized to total PKCα levels 1 × 10^3^ a.u), (*n* = 3 independent experiments; * *p* = 0.0324), (Figure 2A–C). When LC cells are treated with hyposmotic-induced cellular swelling solution, the activity of PKCα, was found to be significantly (*p* < 0.05) increased in glaucoma LC cells versus normal LC cells at the matched time-points of 1, 5, 10, and 30 min, but not at the matched time-points of 2 and 20 min (Figure 2A–C). Exposure of LC cells to hyposmotic solution activates phosphorylation of PKCα in both normal and glaucoma LC cells. The PKCα activation was biphasic and peaked at time-points of 2 min and 10 min. Figure 2D (representative Western blot), and Figure 2E (average data from 3 independent experiments), show a time-course experiment, using time-points where PKCα reached the maximum peak (2 min and 10 min) as shown in Figure 2A,B and the exposure to hypotonic solution was extended to 1 h. We found that the hypotonic cell stretch-induced phosphorylation of PKCα stayed above basal level for at least 1 h of observation (Figure 2D,E).

### 3.2. Basal and Hyposmotic-Induced Expression/Activity of p38-MAPK in Normal and Glaucoma LC Cells p

We then examined the expression/activity of p38-MAPK using a time-course experiment wherein we looked at the expression/activity of p38-MAPK in normal and glaucoma LC cells cultured under resting (isotonic) conditions and exposed to hyposmotic solution. Under physiological (isotonic) conditions, the basal expression level of p38-MAPK protein kinase was significantly elevated in glaucoma LC cells (5.57 ± 0.39, p38-MAPK levels normalized to total p38-MAPK levels 1 × 10^3^ a.u) versus normal controls (3.34 ± 0.28, p38-MAPK levels normalized to total p38-MAPK levels 1× 10^3^ a.u), (*n* = 3 independent experiments; * *p* = 0.0246), (Figure 3A–C). The hyposmotic-induced membrane stretch activates p38-MAPK in a transient and biphasic fashion. The activity of p38-MAPK, measured by its phosphorylation, was significantly (*p* < 0.05) enhanced in glaucoma LC cells at matched time-points of 2, 5, 10, and 20 min, but no significant elevation of p38-MAPK activity was found at a matched time-point of 30 min (Figure 3A–C).

### 3.3. Basal and Hyposmotic-Induced Expression/Activity of p42/44-MAPK in Normal and Glaucoma LC Cells

In the third set of experiments, we used antibodies to site-specific phosphorylated forms of p42/p44 to estimate changes in their expression under physiological (isotonic) conditions, and their activity (phosphorylation) under hyposmotic conditions. Figure 4 shows that under resting (isotonic) conditions, there was significant elevation of p42/44 MAPK phosphorylation in glaucoma LC cells (6.92 ± 0.48, p42/44-MAPK levels normalized to total p42/44-MAPK levels 1 × 10^3^ a.u) versus normal controls (3.13 ± 0.43, p38-MAPK levels normalized to total p42/44-MAPK levels 1 × 10^3^ a.u), (*n* = 3 independent experiments; * *p* < 0.0218), (Figure 4A–C). Treatment of LC cells with the hyposmotic solution induced a significant increase in phosphorylation of p42/p44-MAPK in glaucoma LC cells at the all the matched time-points of 1, 2, 5, 10, 20, and 30 min on average, *p* < 0.05). Similarly to PKCα and p38-MAPK protein kinases, the hyposmotic cell stretch induced activation of p42/44-MAPK in transient and biphasic manner. However, the phosphorylation reached the maximum peak with a decay-time of 5 min for the first peak and 20 min for the second peak, in both normal and glaucoma LC cells (Figure 4A–C).

### 3.4. Expression of PI3K/mTOR Is Elevated in Glaucoma LC Cells

Here, we assessed whether there is any differential expression of additional Ca^2+^-related genes including IP3R, CaMKII, PI3K and mTOR using qRT-PCR. As illustrated in Figure 5, the average data of PI3K, mTOR, IP3R and CaMKII, transcription was detected at low levels in normal control LC cells, but significantly (* *p* < 0.05), up-regulated in glaucoma LC cells. Data are the average of three biological replicates comparing transcription of PI3K, mTOR, IP3R and CaMKII between normal controls to the age-matched glaucoma LC cells.

## 4. Discussion

Today, glaucoma treatments are solely focused onintra-coular pressure (IOP), either by medication or by surgery. It is generally accepted that elevated IOP is the major risk factor for glaucoma, however, it is not necessarily the only cause, as ~25% of patients with glaucoma have no elevated IOP. The reasons for the difficulties of achievement are not completely understood and the idea of clinical therapy is doubtful. Our hypothesis may suggest a new target for combating the optic nerve damage in glaucoma, by interfering with downstream steps such as the PKC, MAPK, and PI3K/mTOR signaling pathways. To date, the association between fibrosis and cell damage in the optic nerve head is not well established.

Human TM [36] and LC [22] cells show elevated cytosolic Ca^2+^, and it is well known that Ca^2+^ is a driver of fibrosis. In order to clarify the molecular mechanisms underpinning fibrosis in glaucoma, we investigated intracellular Ca^2+^ related pathways, for example PKCα and RAS-RAF-MAPK expression and activation in human primary cultures of normal and glaucoma LC cells, using hyposmotic-induced cellular swelling to model glaucoma. We found significantly increased expression (in resting isosmotic solution) and activity (phosphorylation, in hyposmotic solution) of all the three examined protein kinases PKCα, *p*38 and *p*42/44 in glaucoma LC cells. These data may suggest a possible coordinating effect of these protein kinases in the development of fibrosis in glaucoma, and also may provide the molecular bases for the therapeutic outcome targeting PKCα, *p*38MAPK, and *p*42/44MAPK kinases. Hence, a strategy to inhibit their signaling pathways may be crucial for an effective treatment of fibrosis in glaucoma.

PKC family members regulate numerous cellular responses including gene expression, protein secretion, and cell growth and apoptosis. PKCs are implicated in several physiological and pathological mechanisms. It has been found that the PKCα isoform is the most expressed in cardiac myocytes in addition to other PKC isoforms. Interestingly, its expression levels are up-regulated in fetal and neo-natal hearts and decreases with age [37]. It has been also shown that PKCα expression and activity levels are significantly enhanced by cardiac injury indicating that PKCα plays a key role in the progression of cardiac disease [38]. Further studies also found that PKCα expression/activity was enhanced in experimental heart disease including cardiomyocytes hypertrophy, myocardial infarction, ischemia, and heart failure [38,39,40]. PKCα expression and activity has been also reported to be upregulated in a rat model of end-stage heart failure [41]. Deletion of PKCα prevented cardiomyopathy and increased cardiac function in heart failure [42]. Consistent with these studies, for the first time, by utilizing antibodies specific for phosphorylated PKCα, we detected upregulated expression and increased activity (phosphorylation) of PKCα in glaucoma LC cells. The LC cell response to hypotonic stretch by increasing PKCα expression and activity, which was a typical reaction characterized by an initial rapid and transient peak phase at 2 min followed by a second transient peak at 10 min of hyposmotic-induced cell stretch. We showed that normal and glaucoma LC cells responded to hypotonic stretch in a similar way; however, the response to hypotonic stretch was more exaggerated in glaucoma LC cells. The kinetics of the activation of PKCα and MAPKs p38 and *p*42/44-MAPK) by hyposmotic stimulation is biphasic and comprise an early (2 min) and a late (10 min) increase in PKC and MAPK phosphorylation. These data suggest a possible role of PKCα and MAPKs in the development of fibrosis in glaucoma.

Having demonstrated the differential expression and activity of protein kinase PKCα in normal and glaucoma LC cells, we next tested if this is the case in the downstream protein kinases *p*38-MAPK and *p*42/44-MAPK. Likewise, Western blot analysis indicated that basal expression (under isosmotic conditions) *p*38-MAPK and *p*42/44-MAPK increased significantly in glaucoma LC cells, and in response to hyposmotic-induced cellular swelling, the relative activity (phosphorylation) of *p*38-MAPK and *p*42/44-MAPK significantly increased in normal and glaucoma LC cells; however, the response to hyposmotic-induced cell swelling was more exaggerated in glaucoma LC cells for both protein kinases. Interestingly, the response to hypotonic stretch was biphasic and transient, characterized by an initial peak phase at 2 min followed by a second transient peak at 10 min. However, the phosphorylation of *p*42/44-MAPK reached the maximum peak with a decay-time of 5 min for the first peak and 20 min for the second peak for *p*42/44-MAPK in both normal and glaucoma LC cells.

*p*38-MAPK is activated by numerous stress stimuli, including oxidative injury, and osmotic shock [43]. It has been reported that inhibition of *p*38-MAPK, blocked TGFβ-induced collagen expression in fibroblasts, liver, and kidney cells [44]. In another study, blockade of p38-MAPK after the appearance of fibrosis is effective at reducing subsequent renal fibrosis in a model of unilateral ureteral obstruction [45].

In general, the RAS/RAF/MAPK signaling pathway, also known as MEK-1/2/*p*42/44MAPK, is classically stimulated by cell surface receptors including receptor tyrosin kinase (RTK), G-protein coupled receptors (GPCR), and transforming growth factor receptors (EGFR), in a RAS/RAF coordinated sequence [46,47]. However, *p*-38-MAPK pathway is essentially simulated by extracellular stresses such as hyposmotic-induced cell swelling, stretch/strains, and hyper/hypoxia [48,49,50,51]. The MAPK family is protein Ser/Thr kinase, composed by three kinases that include a MAPK, a MAPKK, and a MAPKKK subfamily (Figure 1B). Sequential activation via phosphorylation of the MAPKKKs, induce their interaction with GTP-binding domains of Ras-Rho complex proteins in response to stimuli. Once activated, MAPKKKs phosphorylate and activate MAPKK, which in turn phosphorylate MAPK on Thr-tyr residues located in the loop of the protein kinase domain [27,28,29,30,31,32,33,34,35,36,37,38,39,40,41,42,43,44,45,46,47,48,49,50,51,52], (Figure 1B).

The finding that glaucoma LC cells exhibit a significant increase in expression and activity of PKCα, *p*38-MAPK and *p*42/44-MAPK, prompted us to further investigate other Ca^2+^-dependent key players that may be involved in the production of ECM genes and fibrosis in glaucoma. The PI3K/Akt/mTOR pathway is frequently associated to cellular physiology, including cell metabolism, differentiation, and proliferation. A wide range of diseases such as cancer, cardiac and pulmonary fibrosis, show dysfunction in the PI3K/Akt/mTOR signaling pathways [53]. Changes in this pathway often result in dysfunction of cell proliferation and apoptosis, inflammation, and autophagy, which consequently lead to progression of fibrosis [54]. Therefore, identifying the link between PI3K/Akt/mTOR and *p*38, *p*42/44 signalling pathway and their contribution in ECM gene production and cell proliferation in the lamina cribrosa area will be important in understanding the complex network of these signalling pathways driving the ocular fibrotic process and in finding potential novel therapeutic targets of disease harshness and pathogenesis in fibrotic eye diseases.

In this study, we found elevated levels of mRNA expression level of PI3K, mTOR, IP3R, and CaMKII, in glaucoma LC cells.

Glaucoma is worldwide second leading cause of blindness. The disease is coordinately administrated by large number of factors which drive fibrosis (e.g., TGFβ), and all work via a multitude of downstream signaling pathways including PKCα, MAPKs (*p*38-MAPK and *p*42/44-MAPK), and other downstream signaling such as Rho kinase, JNK, PI-3K/Akt, mTOR, PTEN, and the complex Ca^2+^–calcineurin–NFATc pathways. Comprehension of the key players in these signaling pathways is central for the creation of novel therapeutics which could be of assistance at slowing or preventing the excess of pro-fibrotic ECM gene production in glaucoma. Here, we provide evidence of a possible coordination between PKCα, *p*38-, and *p*42/44-MAPK and/or the downstream PI3K/Akt/mTOR axis (all related to Ca^2+^ signaling) to promote the excess of pro-fibrotic ECM production and LC cell over-proliferation found in glaucoma. Therefore, therapeutics intending to inhibit these pathways in the lamina cribrosa area may halt the development of fibrosis and ONH cupping in glaucoma. Inhibition of certain fibrotic signaling pathways might prevent progressive fibrosis at the optic nerve head, but further studies in clinical models should be performed to prevent ONH cupping in glaucoma.

## Figures and Tables

**Figure 1 jcm-10-00062-f001:**
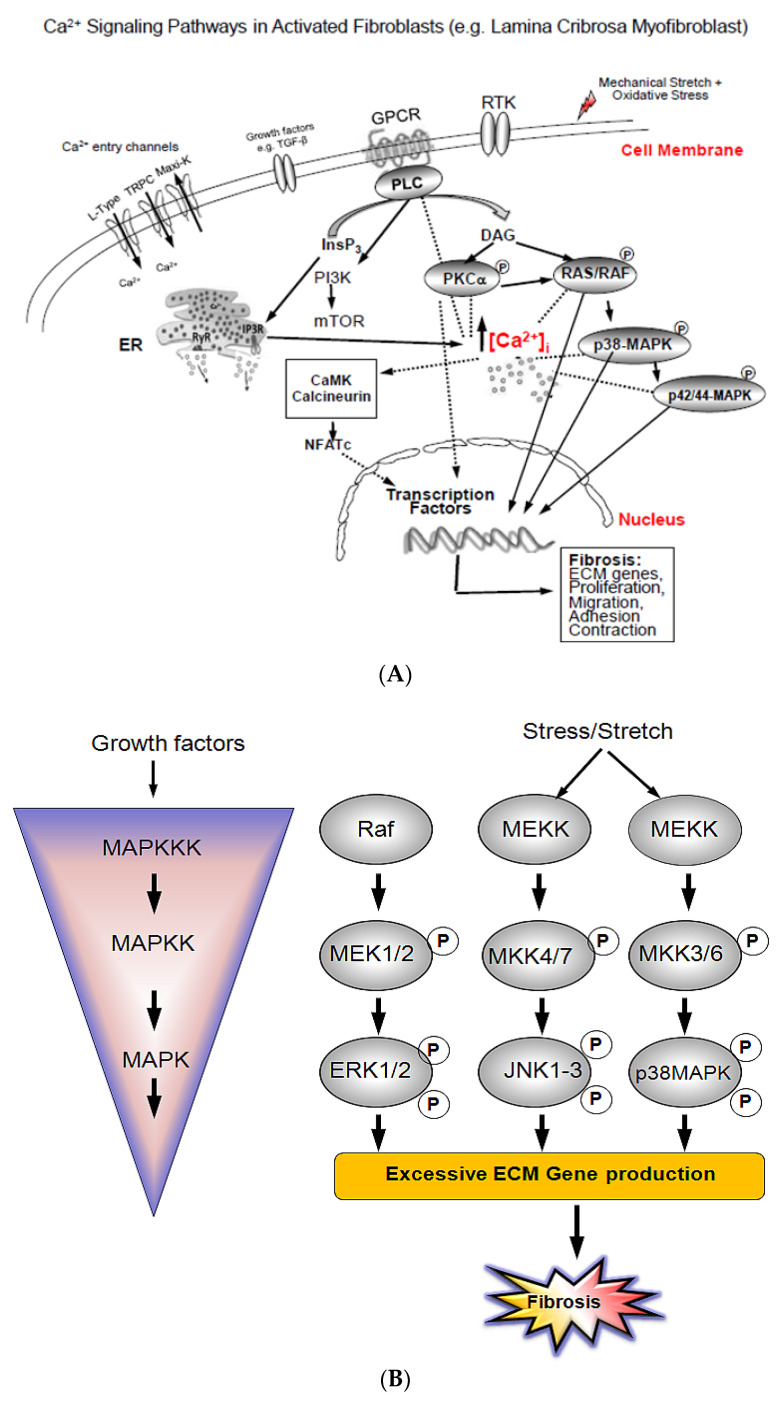
Fibrosis related intracellular Ca^2+^ signaling pathways. (**A**). Schematic mechanism of key pro-fibrotic extracellular matrix (ECM) gene transcription regulation involving different signaling pathways. Including calcineurin/NFATc, PKCα, *p*38-MAPK, and *p*42/44-MAPK in lamina cribrosa (LC) cells. TGF-β, oxidative stress, and/or cell membrane stretch stimuli provoke [Ca^2+^]_i_ movements from Ca^2+^ entry (through TRPC and L-type Ca^2+^ channels) and extrusion of Ca2^+^ from Endoplasmic Reticulum (ER), Ryanodine Receptor (RYR), and inositol triphosphate Receptor (IP3R). [Ca^2+^]_i_ rise activates calmodulin, which in turn activates calcineurin. Activated calcineurin induces dephosphorylation of transcription factors of activated transcription factors of activated T-cells (NFATc), which next translocate to the nucleus to activate Ca^2+^ related genes including those involved in fibrosis in glaucoma. [Ca^2+^]_i_ rise also induce a sequential phosphorylation of PKCα, and downstream series of phosphorylations of *p*38-MAPK, and *p*42/44-MAPK resulting in expression of Ca^2+^-dependent genes such as profibrotic ECM genes and ultimately proliferation of glaucoma LC cells. (**B**) Schematic illustration of the three modules (ERKs, JNKs, *p*38-MAPK). Stimuli, such as cell surface receptors including receptor tyrosin kinase (RTK), growth factor receptors, and G-protein coupled receptors (GPCR), activate up-stream protein kinases, which in turn activate downstream MAPKs. MAPKs signalling pathways are structured in modular cascades in which activation of upstream kinases lead to sequential activation of a MAPK module (MAPKKK, MAPKK, MAPK). MAPKKK phosphorylates (indicated by P) and activates MAPKK, which in turn phosphorylates and activates MAPK. Shown are the major MAPK pathway components, the JNK, and the p38-MAPK modules, and the various cellular responses including fibrosis elicited by MAPK control. MAPK: Mitogen-activated protein kinase; MAPKK: Mitogen-activated protein kinase kinas; MAPKKK: Mitogen-activated protein kinase kinase kinase.

**Figure 2 jcm-10-00062-f002:**
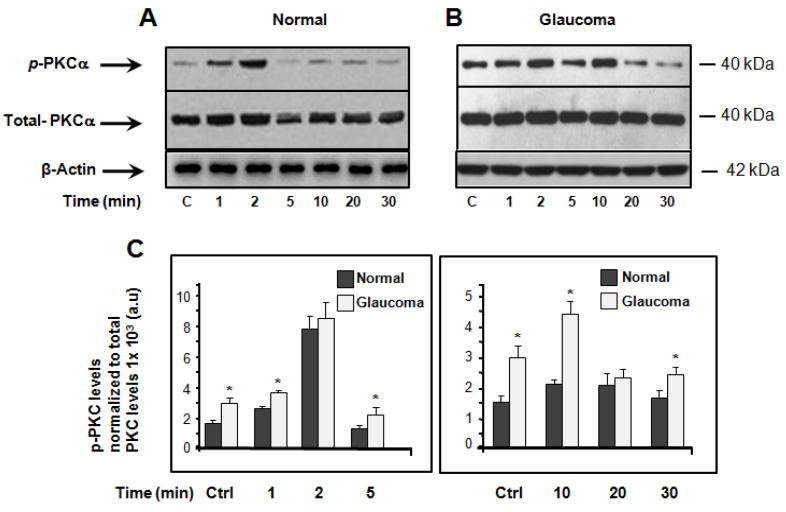
Western blot analysis of total and phospho-specific PKCα in normal and glaucoma LC cells. (**A**,**B**): typical Western immune-blotting analysis of PKCα in LC cell lysates in normal and glaucoma LC cells. Proteins (20 μg/lane) were separated on 10% electrophoresis gels, transferred to nitrocellulose membranes, and immunoblotted with phosphor-and total antibodies specific to PKCα (a 1:1000 dilution), respectively. Time-dependent expression of PKCα in LC cells exposed to isosmotic solution (resting conditions) and activity (phosphorylation) in response to hyposmotic-induced cell swelling. (**C**) Results are averages ± SE of at least 3 biological replicates obtained from three normal and three glaucoma LC cell patients. * *p* < 0.05 compared with time-matched controls (100%). The left panel represents the average data of PKCα expression under basal conditions (Ctrl) and at time-points of 1, 2, and 5 min of LC cells’ exposure to hyposmotic solution. The right panel shows the average data of PKCα expression/activity, under basal (Ctrl) and at time-points of 10, 20, and 30 min of LC cells’ treatment with hypotonic solution. The quantified data are presented in two separate bar graphs, to enhance the difference in PKCα expression/activity between normal and glaucoma LC cells. Note that the scale bar in the left panel (first 5 min) is twice that in the right panel (from 10 to 30 min). (**D**) Representative Western blot analysis of PKCα expression/activity in normal and glaucoma LC cells, illustrating a time-course experiment under isotonic (Ctrl) conditions and after 2, 10, and 60 min of exposure to hyposmotic solution. The time-points chosen here are the time-points where PKCα reached the two transient maximum peaks (2 min and 10 min), and the exposure to hypotonic cell stretch was extended to 60 min. (**E**) Average data ± SE of biological replicates. * *p* < 0.05 compared with time-matched controls (100%). To confirm equal amounts of protein loading, membranes were washed and reblotted with beta-actin antibody (Figure 2A,B).

**Figure 3 jcm-10-00062-f003:**
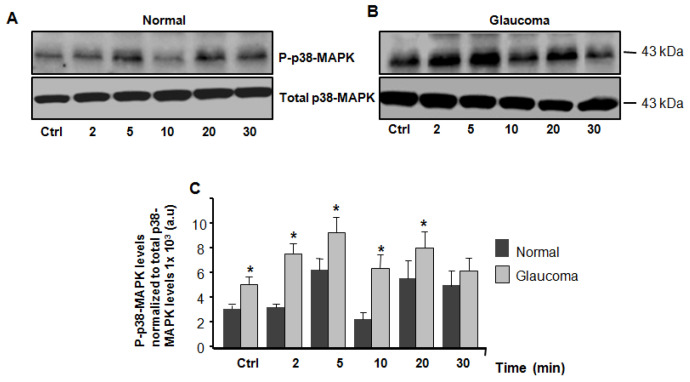
Western blot analysis of total and phospho-*p*38-MAPK in normal and glaucoma LC cells. Equal amounts of cellular proteins (20 μg/lane), were separated on a 10% electrophoresis gel, transferred to a nitrocellulose membrane, and immunoblotted with a 1:1000 dilution of phospho and total antibodies specifically directed against *p*38-MAPK, respectively. (***A***), (Top left panel), representative *p*38-MAPK Western blot of LC cells exposed to isotonic physiological solution (for at least 15 min to obtain a stable baseline) and (B), (Top right panel), LC cells treated with hyposmotic solution at time-points of 2, 5, 10, 20, and 30 min. (**C**) Averages ± SE of at least three biological replicates comparing the expression and activity of *p*38-MAPK in normal and glaucoma LC cells. * *p* < 0.05 was compared with time-matched controls (100%). Average data of expression and activity (phosphprylation) levels of *p*38-MAPK were normalized to total *p*38-MAPK levels. There was a significant increase in phosphorylated *p*38-MAPK (*p*−*p*38-MAPK) protein expression at basal conditions (Ctrl) and *p*38-MAPK activity (phosphorylation) at time-points of 2, 5, and 10 min, but no significant increase in *p*38-MAPK activity was found at 20 and 30 min after exposure to hyposmotic solution.

**Figure 4 jcm-10-00062-f004:**
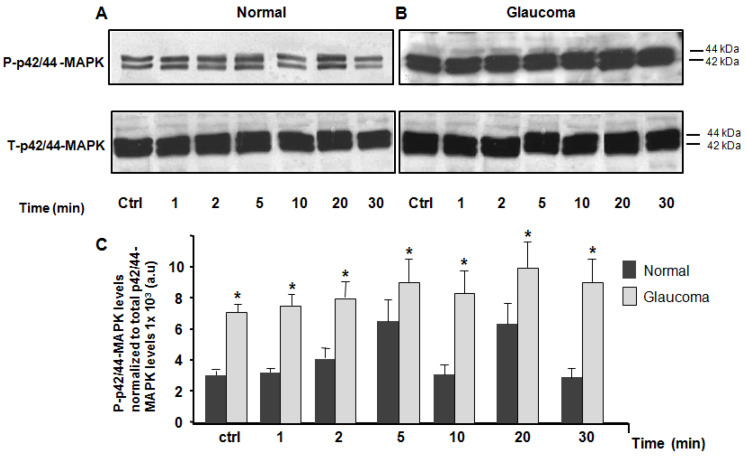
Western blot analysis of total and phospho-*p*42/44-MAPK in normal and glaucoma LC cells. Equal amount of 20 μg/lane of cell protein was loaded, separated on a 10% electrophoresis gel, transferred to a nitrocellulose membrane, and immunoblotted with phospho *p*42/*p*44-MAPK (*p*−*p*42/44 MAPK) and total (*p*−*p*42/44 MAPK) primary antibodies, respectively. (**A**) Top left panel: representative Western blot of LC cells subjected to isotonic physiological solution (for 15 min) and (**B**) Top right panel: representative Western blot of LC cells subjected to hyposmotic-induced cell swelling solution at time-point of 1, 2, 5, 10, 20, and 30 min. (**C**) Averages ± SE of three biological replicates of normal controls and glaucoma LC cells. The average data of expression/activity levels of *p*42/44-MAPK were expressed as a ratio of the total *p*42/44-MAPK levels. Values which are significantly different to controls are denoted by asterisks. * *p* < 0.05 was compared with time-matched controls (100%). Note a significant elevation of phosphorylated hyposmotic-induced cell swelling (*p*−*p*42/44-MAPK) protein expression at basal conditions (Ctrl) and *p*42/44-MAPK activity (phosphorylation) at time-points of 1, 2, 5, and 10 min, 20, and 30 min after exposure to hyposmotic-induced cell swelling solution.

**Figure 5 jcm-10-00062-f005:**
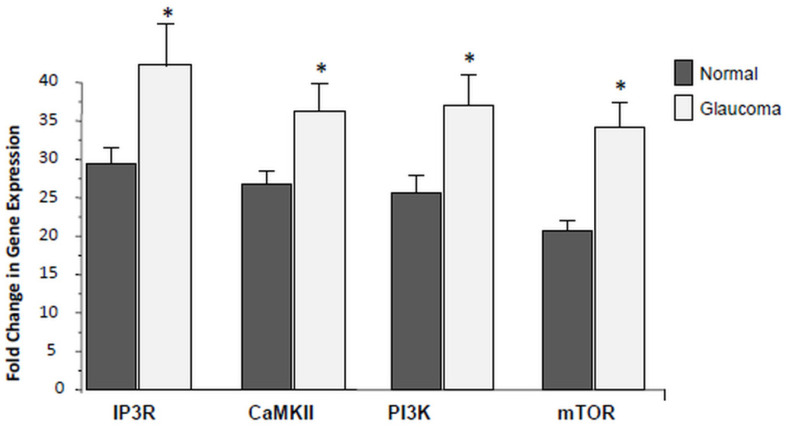
Transcription of IP3R, CaMKII, PI3K, and mTOR in normal and glaucoma LC cells. Average data of qRT-PCR comparing the transcription levels of IP3R, CaMKII, PI3K and mTOR in three 3 normal and 3 glaucoma LC cells. Note that transcription levels of IP3R, CaMKII, PI3K, and mTOR are all significantly up-regulated in glaucoma LC cells. Values which are significantly different to controls are denoted by asterisks (* *p* < 0.05 vs. Control). 18S is the housekeeping control gene. The experiments were performed and reproduced on 3 separate experiments on LC cells from 3 normal controls and from 3 glaucoma donors.

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
