# Peer review of "Intra-Cellular Calcium Signaling Pathways (PKC, RAS/RAF/MAPK, PI3K) in Lamina Cribrosa Cells in Glaucoma"

_jcm, 2020, doi:10.3390/jcm10010062_

Round 1

Reviewer 1 Report

In their manuscript titled “Intra-Cellular Calcium Signaling Pathways (PKC/RAS/RAF/MAPK, PI3K) in Lamina Cribrosa Cells in Glaucoma” Irnaten et al evaluate the intracellular calcium signaling pathways related to fibrosis in normal and glaucomatous lamina cribrosa cells. Overall the manuscript is well written and the results support the conclusions. I have the following questions/concerns:

Methods

Line 142  The sentence starting with “LC cells….is a runon sentence and the authors do not coherently describe the process for hyptonic solution replacement which is obviously important. Please revise.

Results:

Figure 2: The legend for this figure describes panels A and B, but not C, D or E. Please correct.

Figure 2: For quantitative analysis, y-axes in panels C and D, what is meant by “Relative pixel intensity”? Is the intention to represent p-PKC levels normalized to total PKC levels? If so, please clearly state this and label the axes appropriately. Was there a loading control for these western blots?

Figure2: The resolution of the quantification graphs appears to be poor.

Line 229

For the sentence, “We then examined PKC, the expression/activity of p38-MAPK using a time-course…..

This is an awkward transition from PKC to p38, please revise.

In addition, the text of section 3.2 does not refer to the Figure 3 (presumably the figure described). Please make the figure of reference more clear for the reader.

Discussion:

Last sentence “Therefore interventions designed to inhibit these signaling pathways in LC region of the optic nerve head can prevent progressive LC cupping and fibrosis in glaucoma.”

Changes in the pathways discussed in the manuscript are very interesting and add to the field of glaucoma and vision research. However, the data presented do not support this conclusion. Inhibition of certain fibrotic signaling pathways might prevent progressive fibrosis at the optic nerve head, but further studies in models, preclinical work should be performed before drawing this conclusion.

Author Response

"Please see attachment"

Reviewer 2 Report

Connective tissue (lamina cribrosa; LC) in the optic nerve head of the eye is a key site of damage in glaucoma. Excessive extracellular matrix (ECM) deposition leads to remodelling and fibrosis – the result of chronic inflammatory reactions induced by various stimuli, which also cause differentiation of fibroblasts into myofibroblasts, further contributing to remodelling and fibrosis. This may block aqueous humor drainage. The latter transition also involves altered expression of proteins as protein kinase C (PKC) and mitogen-activated protein kinases (MAPK); proteins required in Ca2+ regulation and signalling. LC cells from glaucoma donors present with characteristics of myofibroblasts and express a-smooth muscle actin (a-SMA) and collagen 1A1, periostin and fibronectin upon TGFb stimulation, hypoxia and cyclic stretch. Cytosolic Ca2+ is elevated in human glaucoma LC cells, supposedly via TRPC channels by reducing the oxidative stress-induced production of ECM genes and LC cell proliferation via nuclear factor of activated T-cells (NFATc3) dependent signalling pathway (Irnaten et al., 2018, Invest Opthalmol Vis Sci).

The study by Irnaten and colleagues explores the expression of different kinases involved in Ca2+ signalling pathways in normal and glaucoma LC cells. Irnaten et al. find that cell swelling and resultant membrane stretch brought about by an osmotic challenge activates PKCa and the p42/p44- and p38 MAPK in a transient and biphasic fashion, and suggest a possible coordinating effect of these kinases in the development of fibrosis (in glaucoma), and that the kinases may provide the molecular base for a therapeutic outcome.

This study is of a conceptually important point of interest in regards to identifying a potentially new target for treatment of glaucoma, and supports previous findings of activation of the p38 MAP kinase pathway (albeit in trabecular meshwork cells).

A few important technical and graphical issues decreased enthusiasm for this work though. See comments below.

General

1) The quality of figures throughout the manuscript is bad. In Figure 2C the labelling on the axis is too blurry and not readable. The figure legend in figure 2 is somehow wrong. The panel presents 5 sets of data, but only 2 are mentioned in the legend. Figure 4 is somehow wrong. B is missing in the legend and the blot/membrane is denoted a densitometric analysis. This is in C. “Phosphorylation was induced by HGF”, which is…? Figure 5 is an example of a too small font (see y-axis) which, again, is non-readable. Processing of the graphs/figure panels should be done.

2) Several typos and mixed up sentences are found throughout the manuscript. This should be corrected.

3) There is no such thing as ‘hypotonic cell-membrane stretch’ or ‘hypotonic stretch solution’. It is routinely used, but it is factually wrong. The stretch is not hypotonic and the solution is not hypotonically stretched. A, in this case, hyposmotic challenge is introduced with a resultant stretch of the membrane as a consequence of cellular swelling. The authors should correct the phrasing to ‘hyposmotic-induced cellular swelling’ for example.

4) The overall underlying molecular mechanism is still missing. Upstream from the activation of the kinases many receptors or ion channels could potentially by directly/indirectly involved. The authors should elaborate further on the initial sensory mechanism, which could be stated as a pathology-specific mechanism (although cells in their native settings never encounter an osmotic gradient of that size). In 25% of glaucoma patients IOP is not seen. However, here the routinely used stimulus to induce altered expression pattern or activity of the kinases is an osmotic challenge, which mimics cellular swelling and membrane stretch. Therefore, the authors should discuss the mechanisms behind swelling and Ca2+ influx. Other TRP channels have been suggested to play an essential role in this mechanism.

5) Alongside with this the authors could experimentally try to address an upstream target and not only rely on the kinases in regards to therapeutic treatment.

Methods

6) In the method section (Western blotting) it is written that 20ug protein is loaded in each lane. In the main text/figure legend it says 10ug? Please align and correct to the amount actually loaded.

6i) Please state the size (kDa) of the bands detected in all blots (in all panels) as in Figure 2D.

7) For how long was the LC cells treated with hyposmotic solution prior to RNA extraction? From 1-30 min? Were there any controls of the overall viability or tracking of the state of cells during the time course?  

Results

Page 6, line 207

“…when LC cells are exposed to hypotonic cell-membrane 207 stretch, the activity of PKCα, was found to be significantly (p<0.05) increased in glaucoma LC cells 208 at all the indicated time points of exposure to hypotonic stretch (Figure.2 A, B and C).”

Is that really the case after 2 min or 30 min – if one look at the total expression? It is not clear to why the quantified data are presented in two separate bar graphs in Fig.2C, and the comparison between the ctrl group and LC cells at 2 min does not look statistically significant different. Stats/asterisks are missing in the figure panel.

Interesting with the peaks at time 2 min and 10 min. Please discuss the underlying mechanism for this peak expression at these time points/decay times. The authors briefly touch upon this in the discussion (line 315). Please elaborate.

Page 7, line 229

“We then examined PKCα, the expression/activity of p38-MAPK using…”. The experimenters investigate the p38-MAPK expression and activity, I believe?

Figure 3, the authors should state if A) or B) is LC cells or normal cells. It seems one of the bars in the graph (C) has been sliding down the x-axis…?

Did the authors think about running a time-course with expression peaks as for PKCα?

Page 8, line 235

“Total p38 MAPK protein phosphorylation was not different under basal or hypotonic cell stretch condition.” So, quantifying the level of total protein at time point 5 and 10 min is not significantly different (in the normal cells)? Please state the p values as for the other comparisons. This is also the case for total levels of PKCα after 1 and 2 min, respectively, it seems.

Page 8 line 253

“…there was significant (2.7-fold on average, p < 0.05) higher level of phosphorylation of p42/p44-MAPK in glaucoma (similar to PKCα and p38-MAPK)…”

For PKCα this difference was around 0.3-fold at 2 min and 2.5-fold at 10 min, but not stated at resting level. For p38-MAPK the difference was 1.9-fold. Stating that LC cells and normal cells are similarly elevated at resting levels is fairly overstated.

Page 9, line 272

Clinically, it seems a very low number of patients/donors (3) included. Does this number adhere to the requirements of the journal?
